# High Blood Pressure and Impaired Brain Health: Investigating the Neuroprotective Potential of Magnesium

**DOI:** 10.3390/ijms252211859

**Published:** 2024-11-05

**Authors:** Khawlah Alateeq, Erin I. Walsh, Nicolas Cherbuin

**Affiliations:** 1National Centre for Epidemiology and Population Health, Australian National University, Canberra, ACT 2601, Australia; khawlah.alateeq@anu.edu.au (K.A.); erin.walsh@anu.edu.au (E.I.W.); 2Radiological Science, College of Applied Medical Science, King Saud University, Riyadh 11451, Saudi Arabia

**Keywords:** blood pressure, magnesium (Mg), brain ageing, dementia

## Abstract

High blood pressure (BP) is a significant contributor to the disease burden globally and is emerging as an important cause of morbidity and mortality in the young as well as the old. The well-established impact of high BP on neurodegeneration, cognitive impairment, and dementia is widely acknowledged. However, the influence of BP across its full range remains unclear. This review aims to explore in more detail the effects of BP levels on neurodegeneration, cognitive function, and dementia. Moreover, given the pressing need to identify strategies to reduce BP levels, particular attention is placed on reviewing the role of magnesium (Mg) in ageing and its capacity to lower BP levels, and therefore potentially promote brain health. Overall, the review aims to provide a comprehensive synthesis of the evidence linking BP, Mg and brain health. It is hoped that these insights will inform the development of cost-effective and scalable interventions to protect brain health in the ageing population.

## 1. Introduction

The number of individuals aged 65 and over is projected to more than double globally in the next decades, increasing from ~0.7 billion in 2019 to ~1.5 billion in 2050 [1]. This will have significant implications for the prevalence of age-related diseases and the corresponding burden of disease [2]. Dementia in particular poses a substantial challenge as there are currently no effective treatments available. However, accumulating evidence suggests that modifiable risk factors play a crucial role in the underlying pathological processes and may be amenable to prevention and risk reduction strategies [3]. Although several modifiable risk factors have been identified [4], a more precise understanding of their time course, underlying biological mechanisms, and the extent of their impact is necessary.

This review focuses specifically on blood pressure (BP), which is known to be a major contributor to cerebrovascular disease [5], neurodegeneration [6,7,8], cognitive impairment [9,10], and dementia [11,12,13]. High BP is the leading risk factor for premature death globally [14].

A large body of knowledge characterizing the role of high BP in cardiovascular disease is available; however, a comprehensive understanding of its role in brain ageing requires further investigation. Given the urgent need to identify strategies that effectively reduce BP, which are affordable and scalable at the population level, and minimize its detrimental effects on brain health, this review will also focus on magnesium (Mg), particularly dietary Mg, as its antihypertensive properties are well-established [15,16,17]. Importantly, when considered in the context of dietary intake, Mg has the potential to contribute to highly scalable interventions in the population.

To contextualize the pathophysiological effects of BP and the protective effects of Mg, we first briefly discuss the major ageing mechanisms that are likely to be implicated in the pathological processes underlying the effects of BP in neurodegeneration and cognitive decline. We then summarize the typical brain and cognitive changes that occur with ageing. The second part of this review then covers the epidemiology, measurement, mechanisms, and impact of BP on brain ageing and cognitive decline. Finally, the third section focuses on the contribution of Mg to the ageing processes, including its contribution to lowering BP and its influence on brain health.

## 2. Ageing

This section provides a brief overview of major physiological mechanisms underlying the ageing process and the degree to which they contribute to brain ageing, and cognitive decline.

### 2.1. Ageing Mechanisms

The ageing mechanisms can be broadly categorized into four main types: accumulation of cellular damage linked to oxidative stress (OS) and inflammation, increasing genomic instability, loss of proteostasis, and telomere shortening. OS arises when there is an imbalance in the body’s capacity to neutralize reactive oxygen species (ROS), which are natural by-products of cellular metabolism. Excessive production or insufficient buffering of ROS results in increased OS, which in turn leads to the release of cytokines and the development of a chronic pro-inflammatory state often referred to as “inflammaging” [18,19]. Chronic low-grade inflammation, which involves a dysregulation of the immune system and a low-grade but persistent increase in pro-inflammatory factors, contributes to accelerated ageing [20]. This state promotes genomic instability, cellular senescence, apoptosis, and ultimately contributes to biological ageing [18,19,21]. As organisms age, genetic alterations accumulate, affecting cellular and tissue functions and impacting health [22]. DNA damage, caused by free radicals (ROS that have not been buffered by anti-oxidants) and inadequate repair mechanisms, disrupts gene regulation, leading to cell death or senescence [23,24,25]. Proteostasis, the maintenance of protein balance through synthesis, folding, trafficking, and degradation, is crucial for cellular function [26], and is disrupted by OS. The consequent accumulation of misfolded proteins and aggregates impairs cellular function and contributes to age-related decline and neurodegenerative diseases such as dementia [27]. Telomeres, the protective DNA caps at chromosome ends, maintain genome integrity and as they shorten with each cell division, their shrinkage is a hallmark of biological ageing [28,29]. OS, environmental (e.g., smoking), health (e.g., hypertension, and obesity), and poor diet are factors known to accelerate this shortening, and impact cellular health and the ageing process [25]. Moreover, shortened telomeres lead to replicative senescence, causing tissue and organ decline [30].

### 2.2. Brain Ageing

The physiological changes associated with ageing occur throughout the body, and in the brain result in neurodegeneration. The following sections summarize how the age-related changes reviewed above impact the brain structure and function.

#### 2.2.1. Age-Related Microscopic Changes

Age-related processes are closely linked to important microscopic changes within the brain. Elevated OS and chronic inflammation (inflammaging) play pivotal roles in the loss of neurons, glial cells, and myelin. In the brain, OS damages neurons, triggers microglial activation, and releases pro-inflammatory cytokines including interleukin (IL)-1β, IL-6, IL-17, IL-18, and Tumor Necrosis Factor-*α* (TNF-α) [20,31]. This cascade further promotes chronic inflammation and OS, resulting in neuronal death, loss of dendritic spines, dendritic tree atrophy, and shrinkage of the neuropil [18,19,21]. These inflammatory processes also compromise the integrity of the axons’ myelin sheaths within white matter (WM) tracts [32,33] and the maturation of oligodendrocyte precursor cells responsible for its replacement [34]. This progressive demyelination [35,36] results in Wallerian degeneration, synaptic dysfunction, and compromised brain connectivity [37,38].

Additionally, OS-related DNA damage promotes neuronal dysfunction, and impaired synaptic communication, while also hindering neurogenesis [39,40]. This structural and functional damage is further aggravated by the accumulation of misfolded proteins and aggregates, such as tau inside and amyloid beta (Aβ) around neurons, particularly in the hippocampus, a vital region for memory and learning [27,41,42]. As these proteins are neurotoxic, they interfere with normal neural function, and contribute to synaptic loss, diminished connectivity, and ultimately neuronal death [43,44,45].

In addition to cellular damage, cerebral microvasculature dysfunction contributes to impaired brain health. Animal studies show that elevated OS can compromise microvessel endothelial function and disrupt the blood–brain barrier (BBB) integrity [46,47,48,49]. When the BBB becomes compromised, leakage of fluids, proteins, and plasma into perivascular brain tissue triggers pro-inflammatory processes, impairs vasodilation, disrupts blood flow, and promotes chronic hypo-perfusion, brain oedema, and arterial stiffening. This cascade contributes to secondary neurodegeneration [50], impedes the maturation of oligodendrocyte precursor cells, and disrupts myelin repair [51].

#### 2.2.2. Age-Related Macroscopic Changes

The accumulation of microscopic changes discussed above progressively leads to the development of macroscopic changes across the brain [52]. At the population level the average total brain volume shrinks relatively linearly between the ages of 20 and 80 years at an rate of ~0.4% per year [53,54,55]. Simultaneously, ventricles tend to enlarge, at an average annual rate of ~1.8% [53,56]. However, the extent of these changes varies across different brain regions and between individuals.

Gray matter (GM) loss contributes most to brain shrinkage and its volume declines [54,55,57] at an average rate of 3.8% per decade between the ages of 20 and 70 [57] and more so in women [58]. The most vulnerable GM region is the hippocampus, which shrinks by approximately 2% by age 40, 8% by age 60, and up to 20% by age 80 in cognitively normal individuals [59]. Moreover, the annual rate of hippocampal atrophy is higher in mild cognitive impairment (MCI) (2.53% per year) [60], and even more so in Alzheimer’s disease (AD) (4.66% per year) [61].

In contrast to GM, WM volume tends to remain relatively stable during young adulthood and reaches its peak in the mid-40s [62,63]. WM volume declines by approximately 0.39% per decade for those in their 40s, to 0.61% per decade for those in their 80s [53], and more so in men than women [58]. One major contributor to WM loss is attributable to the development of white matter lesions (WMLs), particularly in the subcortical WM [64,65]. WMLs are generally small and infrequent in younger individuals but increase notably with age, especially in those above 55 years [66]. with annual increase rates varying between 4.4% and 37.2% per year [67]. Together, these macroscopic changes in the brain contribute to a loss of function, progressive cognitive decline, and the development of dementia, which are discussed next.

### 2.3. Cognitive Decline

Age-related micro- and macroscopic changes in the brain structure impact cognitive functions and cause cognitive decline and the progression of neurological disorders, including dementia. Cognitive functions undergo rapid improvement during development [68,69]. As individuals transition into mid to late adulthood, most cognitive functions exhibit a somewhat linear decline [70,71]. For example, average reaction time decreases by approximately 15% between the early 20s and early 40s [72]. In order to develop strategies to slow down or prevent cognitive decline, it is important to consider the factors that contribute to changes in cognitive function as individuals age, which is the focus of the next section.

### 2.4. Factors Contributing to Brain Ageing and Cognitive Decline

Brain ageing and cognitive decline are influenced by genetic and environmental factors. Genetics contribute to approximately 30% to 60% of the observed differences in cognitive decline [73]. The remaining 40–70% is explained by modifiable environmental, health, and lifestyle factors—such as low education, obesity, diabetes, cardiovascular disease, lack of physical activity, and smoking [3,70].

Among these factors, cardiovascular disease is a major contributor to cognitive decline and dementia, and BP is often used as an index of cardiovascular health. High BP has been associated with increased Aβ deposition, neuroinflammation, WMLs, and brain shrinkage [74,75,76], which are known to contribute to brain ageing and cognitive decline [9]. However, our understanding of when and to what extent high BP starts to influence brain and cognitive functions remains incomplete. The next section reviews our current understanding of these questions and identifies important knowledge gaps.

## 3. BP and Ageing

### 3.1. Definition

BP measures the force of blood against vessel walls during both systole and diastole, andis expressed in millimeters of mercury (mmHg) [77]. There is incomplete agreement as to what constitutes normal BP. The American Heart Association (AHA) defines normal BP as a systolic BP (SBP) < 120 mmHg and a diastolic BP (DBP) < 80 mmHg, in the absence of any antihypertensive medication use. Elevated BP is SBP from 120 to 129 mmHg and DBP < 80 mmHg. Hypertension stage 1 is characterised by SBP ranging from 130 to 139 mmHg, and/or DBP ranging from 80 to 89 mmHg. Hypertension stage 2 is generally defined as SBP ≥ 140 mmHg and/or DBP ≥ 90 mmHg [78]. However, guidelines differ. For example, the European Society of Cardiology/European Society of Hypertension, (ESC/ESH) defines optimal BP as a SBP < 120 mmHg and a DBP < 80 mmHg, in the absence of medication; Normal BP as SBP between 120 and 129 mmHg and/or DBP between 80 and 84 mmHg; with high normal BP being characterised by SBP between 130 and 139 mmHg, and/or DBP between 85 and 89 mmHg; and hypertension as SBP ≥ 140 mmHg and/or DBP ≥ 90 mmHg [79].

Additionally, two other measures, mean arterial pressure (MAP) and pulse pressure (PP), also provide important information about cardiovascular health. MAP represents the average pressure within the arteries during a cardiac cycle, computed as (SPB + 2 * DBP)/3, reflecting the perfusion pressure in organs and tissues [80]. PP, on the other hand, is the difference between SBP and DBP, and provides information about arterial wall force and stiffness [81].

### 3.2. BP Regulation

BP regulation involves both central and peripheral mechanisms to maintain optimal blood flow and a stable BP.

#### 3.2.1. Central BP Regulation

BP is centrally regulated in the medulla oblongata, which modulates blood vessel contractility and heart function [82,83] via the sympathetic (SNS) and parasympathetic (PNS) nervous systems [84,85]. The SNS increases BP by releasing norepinephrine and epinephrine, while the PNS lowers BP through acetylcholine release, promoting vasodilation and reduced heart rate. The balance between SNS and PNS activity maintains BP homeostasis in response to various factors [85,86].

#### 3.2.2. Peripheral BP Regulation

Peripheral BP regulation involves mechanisms that adjust blood vessel contractility and heart activity [87]. A key component is the baroreceptor reflex, where sensors in the aortic arch and carotid arteries respond to BP changes and signal the medulla oblongata [88]. High BP triggers a decrease in heart rate and vasodilation, while low BP prompts an increase in heart rate and vasoconstriction [89,90].

Another mechanism is the renin-angiotensin-aldosterone system (RAAS), which regulates BP and fluid balance. When BP drops, renin converts angiotensinogen into angiotensin I, which is then converted into angiotensin II, a vasoconstrictor that raises BP and stimulates aldosterone release to retain sodium and water, further increasing BP [91,92].

In addition, the endothelium, the inner lining of blood vessels, also contributes to BP regulation by releasing chemokines that affect vascular tone [87]. Activation of the parasympathetic nervous system triggers the release of nitric oxide (NO), a vasodilator that relaxes blood vessels and lowers BP [93,94]. The endothelium also interacts with the RAAS, releasing renin to increase BP in response to low BP [95,96,97]. Thus, endothelial dysfunction can lead to persistent BP elevation and increased cardiovascular disease risk, as discussed in the next section [98].

Central and peripheral BP regulation systems are closely linked to the brain’s adaptive mechanisms, which ensure optimal cerebral blood flow [99,100,101,102]. These adaptations involve structural changes in cerebral arteries and arterioles, such as increased wall-to-lumen ratio, to reduce vascular stress [103]. This remodelling includes alterations in cell growth, migration, and extracellular matrix composition [5].

Additionally, adaptations to high BP include enhanced myogenic constriction in cerebral arteries and arterioles, which helps protect delicate microvessels from damage [103,104]. This constriction also plays a role in autoregulation, maintaining stable cerebral blood flow during BP fluctuations [103,105].

### 3.3. Effect of Sex Hormones

Sex hormones have an important influence on the cardiovascular system, and in the regulation of BP levels. For example, oestrogen particularly influence BP levels in women by reducing renin release, which lowers angiotensin II production. This prevents excessive vasoconstriction, helping to maintain lower BP in premenopausal women [106,107]. Conversely, reduced oestrogen levels during menopause are associated with elevated BP. Research has consistently shown that postmenopausal women who undergo significant reduction in oestrogen levels are at a higher risk of developing hypertension, with prevalence rates ranging from 36.76% to 44.1% [108,109,110]. Hence, the impact of sex on BP levels is of particular relevance, underscoring the importance of accounting for sex-related factors when assessing the effects of BP on health during the ageing process.

### 3.4. Measurement

Accurate BP measurement is an important tool in the assessment of cardiovascular health and hypertension management. It can be measured invasively via central or peripheral catheters, or non-invasively with sphygmomanometers on the arm or forearm. Central BP is more precise but less commonly used due to its invasive nature [111,112].

Central and peripheral BP measurements can differ, with brachial artery SBP being typically higher than central aortic SBP due to the amplification phenomenon. This difference decreases with age as central arteries stiffen more rapidly than peripheral ones [113,114]. Peripheral BP can be measured manually with a cuff and stethoscope, which is prone to observer errors [115,116], or with automated devices using the oscillometric method, offering standardized measurements with less observer bias [111]. Automated devices, while slightly less precise, are preferred for their consistency [117].

BP can be measured in clinics or at home. Clinical measurements may be affected by “white coat hypertension”, while home monitoring with validated devices helps track BP over time and reduce the white coat effect [118]. The AHA recommends home monitoring for reliable BP tracking [117].

### 3.5. Epidemiology

Epidemiological studies have consistently shown that different BP components undergo distinct changes throughout an individual’s lifespan. The trajectories of SBP and DBP exhibit a gradual increase from adolescence to adulthood [119,120]. This increase tends to accelerate with advancing age until the age of 50 years [120,121,122,123]. The rise in both SBP and DBP is largely attributable to the age-related changes in peripheral vascular resistance [120]. DBP typically reaches a peak in the 50s, and then slowly decreases from the age of 60s into old age [121,124]. In contrast, SBP continues to increase beyond 50 years [120,125], mostly due to ongoing arterial calcification and increased stiffness [119].

These age-related patterns in DBP and SBP contribute to significant increases in PP and MAP over time [81,126,127]. This is particularly important because MAP is the main determinant of blood flow in small blood vessels, which are particularly vulnerable in organs such as the kidneys and the brain [128,129].

The increase in SBP/DBP contributes to the development of clinical hypertension, which is a highly prevalent condition that affects 1.27 billion individuals worldwide [130]. However, hypertension prevalence varies by country, with higher rates in low- and middle-income countries (31.5%) compared to high-income countries (28.5%) [131]. Ethnic disparities are evident, with higher rates in Central and Eastern Europe (women 77%, men 63%), Latin America and the Caribbean (women 72%, men 57%), and Central Asia, the Middle East, and North Africa (women 64%, men 47%) [130]. Hypertension prevalence also varies by generation, age, and sex, with onset occurring at younger ages in newer generations and exceeding 20% among older adolescents and young adults [132]. This trend highlights the need to closely examine the impact of rising BP in younger individuals. Hypertension diagnosed in late adolescence and young adulthood poses a long-term risk for cardiovascular and cerebrovascular events and should be managed with the same urgency as in older populations (Figure 1) [132].

Contrary to the belief that men generally have higher BP, studies show this is primarily true at younger ages [124,133]. As individuals age, women tend to have higher BP, starting as early as their thirties [120,133], and increasing further after menopause [108,109,110], leading to a narrowing of the BP disparity between the sexes over time [134]. Importantly, a recent study suggests that a higher number of women develop hypertension compared to men after the age of 64 years [135], possibly due to the decline in oestrogen levels observed with advancing age [136].

These epidemiological changes necessitate comprehensive research to evaluate potential health implications effectively. Moreover, these epidemiological shifts are likely influenced by changes in lifestyle, diet, and other environmental factors [130,137]. Consequently, it is essential to consider the contribution of these risk factors when assessing the impact of rising BP on health.

### 3.6. Risk Factors

Several genetic and environmental risk factors contribute to increasing BP and the development of hypertension. Genetics contribute approximately 30% to 50% in BP variability in all stages of life [138,139,140,141]. Environmental factors including diet, obesity, physical inactivity, excessive alcohol consumption, and smoking, as well as others, account for the remaining variability in BP.

Diet quality is associated with an increased risk of hypertension in several ways. Increasing sodium intake in food from 3.5 g/day to 10.5 g/day is associated with increased SBP (+4.3 mmHg) and DBP (+2.3 mmHg) [142]. Moreover, diets rich in saturated fats elevate the risk of developing hypertension by about 12% [143]. A substantial body of evidence also indicates that insufficient intake of potassium and Mg is associated with increased BP levels [144,145]. Alcohol is another dietary component that contributes to elevated BP levels. Robust evidence indicates that high alcohol intake (>30 g/day) results in increased SBP (+3.7 mmHg) and DBP (+2.4 mmHg) [146]. Interestingly, alcohol consumption can lead to Mg deficiency, potentially contributing to elevated BP [147]. Ethanol, found in alcoholic beverages, acts as a Mg diuretic, leading to Mg excretion and gradual depletion with excessive and prolonged consumption [148].

In addition to diet quality, excessive energy intake also increases the risk of hypertension. Being overweight or obese and leading a sedentary lifestyle are significantly associated with elevated BP and hypertension. Around 40% of hypertension cases can be attributed to obesity [149]; however, this effect appears to be partly reversible. A meta-analysis of 25 randomized controlled trials (RCT) involving 4874 participants demonstrated that interventions combining calorie restriction and increased physical activity, resulted in an average weight loss of 5.1 kg, and importantly led to significant reductions in SBP (−4.44 mmHg) and DBP (−3.57 mmHg) [150]. Even light exercise such as walking positively influences BP and reduces hypertension risk [151]. Another comprehensive meta-analysis of 17 studies with 12,046 participants indicated that moderate-intensity leisure time physical activities including walking, running, bicycling, and soccer lowered SBP (by an average of −5.35 mmHg) and DBP (by an average of −4.76 mmHg) compared to a control group [152].

Finally, smoking is a major, completely preventable risk factor for hypertension. A follow-up of 13,529 men (mean age 51.2 years) over 14.5 years showed a clear link between past/current smoking and an 8% and 15% higher risk of hypertension, respectively, compared to non-smokers [153]. In the Women’s Health Study (n = 28,236 women), smoking exhibited a clear association with hypertension risk. Former smokers (1–14 cigarettes/day) had a 3%, current smokers (1–14 cigarettes/day) a 2%, and those smoking ≥15 cigarettes/day had an 11% increased risk of hypertension compared to never smokers, after adjusting for multiple factors [154]. Unlike genetics, these factors are amenable to modification, offering an opportunity for intervention and lifestyle adjustments that can positively impact BP levels and overall cardiovascular health.

### 3.7. BP-Related Health Conditions

High BP often coexists with chronic conditions such as type 2 diabetes [155], cardiovascular disease [156], chronic kidney disease (CKD) [157,158], and cerebrovascular disease [159]. Furthermore, hypertension is a risk factor for these conditions and significantly increases the risk of CKD and end-stage renal disease (ESRD), with risk increasing continuously above 120 mmHg [156,157]. This connection highlights the importance of BP-lowering interventions in reducing overall disease burden.

### 3.8. BP Treatment

Antihypertensive treatment includes non-pharmacological and pharmacological approaches. Non-pharmacological interventions include dietary modifications such as increasing Mg intake. Mg helps relax blood vessels, leading to a reduction in BP [160]. Studies have shown that increasing dietary Mg, either through supplements or Mg-rich foods, can contribute to lowering SBP and DBP [161]. This approach, combined with regular exercise, and weight loss, contributes to a comprehensive strategy for managing hypertension [162,163,164]. When non-pharmacological interventions are ineffective, pharmacological interventions are recommended [165]. Indeed, medication plays a major role in reducing the risk of cardiovascular disease and premature mortality in the population. In a meta-analysis of six trials with 27,414 participants (mean age: 70 years; 56.3% female), intensive BP treatment, targeting SBP below 140 mmHg, reduced major cardiovascular events by 21% for individuals with hypertension aged 60 years and older [166]. It demonstrated that reducing SBP by 10 mmHg resulted in a decreased risk of 20% for major cardiovascular events, 17% for ischemic heart disease, 27% for stroke, 28% for heart failure, and 13% for all-cause mortality [167].

Thus, the prevention, treatment, and management of hypertension hold significant implications for a number of diseases. However, what remains uncertain is whether antihypertensive treatment is the most effective and desirable approach to mitigate the impacts of the progressive increase in BP observed in ageing. Thus, the next section explores the links between BP, antihypertensive treatment, and pathological ageing.

### 3.9. BP and Ageing Mechanisms

As individuals age, progressive endothelial damage leads to endothelial dysfunction characterised by a decreased capacity for the vascular wall to relax due to decreased availability of NO, and increased endothelium-derived contracting factors [168,169]. Research in hypertensive rat models has demonstrated that increased OS deactivates NO, impairs NO synthase isoforms, and further disrupts endothelial function [170], leading to a reduction in the diameter of blood vessels and subsequently, an increase in BP levels [171,172]. 

In addition to localised effects, increased systemic inflammation is also linked to higher BP levels. Consistent human findings demonstrate that elevated plasma levels of pro-inflammatory cytokines, including IL-6 and C-reactive protein (CRP), in middle-aged and older individuals without cardiovascular disease are associated with an increased likelihood of developing hypertension [173]. Moreover, animal studies have shown that proinflammatory cytokines, including IL-1β [174], TNF-α, IL-6 [175,176], and interferon-gamma (IFN-γ) [177], are elevated in hypertensive animals, and that this cytokine release leads to increased BP through enhanced sympathetic nerve activity. In contrast, mice lacking certain proteins such as IL-6 [178], TNF-α [179], or IL-17 [180] exhibit lower BP when exposed to a hypertensive dose of Angiotensin II (powerful vasoconstrictor) compared to controls. Similar effects have also been shown in humans [181,182] and demonstrated positive associations between higher serum TNF-α concentration, higher systolic BP, and the development of hypertension [183,184,185].

Inflammation is also associated with atherosclerosis, a process in which fatty deposits, cholesterol, and other substances accumulate and gradually form plaques that narrow and harden the arteries, reducing blood flow and increasing BP [186,187]. Vascular calcification, which involves the build-up of calcium (Ca) deposits within blood vessel walls, increases with age. This calcification reduces blood vessel flexibility and contributes to higher BP and hypertension [188]. Increased OS linked to endothelial dysfunction can initiate and advance the development of atherosclerosis by oxidizing circulating low-density lipoproteins (LDL), making them more prone to uptake by immune cells, especially macrophages in arterial walls [189,190]. Increased production of OS and inflammation also contribute to vascular calcification [186,191,192], by promoting the transformation of vascular smooth muscle cells (VSMCs) into osteoblast-like cells [193], which contribute to the calcification process [194]. This further impairs endothelial function and further contributes to raising BP.

Impairment of central BP control mechanisms also contributes to accelerating the development of hypertension. Indeed, increasing BP levels and other related pro-inflammatory mechanisms contribute to neurodegeneration, which also affects the BP regulatory system and leads to overstimulation of the vasomotor centre. This produces extended vasoconstriction and increased blood vessel resistance, resulting in sustained high BP [82]. Moreover, high BP also leads to an overactive sympathetic system, which raises heart rate and contractility, and thus diminishes the effectiveness of the parasympathetic nervous system in countering the action of sympathetic effects [85,86]. This leads to chronically high BP and eventually hypertension due to persistent vasoconstriction, increased heart rate and altered vascular tone [101,102], which compounds the effects of endothelial dysfunction, atherosclerosis, vascular remodelling, and arterial stiffness [99].

In addition to systemic effects, the disruption of the BP regulatory system has a profound impact on the adaptation mechanisms of the brain [99,100,101,102]. Over time, elevated BP can lead to vascular remodelling in cerebral arteries and arterioles, resulting in increased stiffness [35]. These changes lead to significant microscopic alterations in the brain, impacting the BBB and promoting small vessel disease and neurodegeneration, detailed in the next section.

### 3.10. BP and Brain Ageing

The previous section discussed the main mechanisms contributing to peripheral and cerebrovascular damage, which frequently develop with ageing. This section summarises the microscopic and macroscopic brain changes associated with elevated BP (Figure 2).

#### 3.10.1. BP-Related Microscopic Changes

High BP and hypertension are linked to neuronal dysfunction and microscopic alterations in the brain microenvironment [105,195,196]. This strong association between high BP, OS, and inflammation exerts detrimental effects on both neurons and glial cells. Experimental models of hypertension consistently show that hypertension activates the microglia, increases the production of pro-inflammatory cytokines, and increases OS in critical brain regions, including the hippocampus, hypothalamus, amygdala, and stratum [197,198,199,199,200]. This pro-inflammatory environment is known to contribute to neuronal damage, glial and dendritic loss, potentially leading to neurodegenerative diseases [18,19,21]. Neuroinflammation in hypertension also leads to myelin disruption in white matter tracts [38] as demonstrated by altered diffusion imaging parameters in hypertensive individuals [201].

High BP impairs neurogenesis. Mice induced with hypertension through a four-week angiotensin II infusion, displayed reduced synaptic density in the hippocampus’s stratum radiatum [202]. Additionally, hypertension is associated with increased misfolded protein accumulation, evidenced by Aβ deposits in hypertensive mice just four weeks after induction [203]. Furthermore, hypertension promotes tauopathy independently of Aβ, potentially through oxidative damage and increased brain inflammation [204].

In addition to the neuronal dysfunction, high BP may lead to microscopic changes by affecting the cerebral micro–vasculature [205]. Elevated BP induces structural and functional changes in microvessels, leading to vascular remodelling that impairs cerebral blood flow, disrupts nutrient delivery, and compromises waste removal, contributing to neuronal and glial cell stress and injury [205].

A key consequence of high BP-induced microvascular damage is the disruption of the BBB, consequently contributing to neurodegeneration [35,103]. High BP disrupts the integrity of the BBB by inducing endothelial dysfunction, promoting oxidative stress, and enhancing inflammatory responses [205]. This disruption allows neurotoxic proteins and pro-inflammatory molecules, such as fibrinogen, to enter the brain parenchyma [35,103,205], triggering inflammation, cerebral oedema, demyelination, axonal damage and reduced fibre density [50,206]. Moreover, hypertension also increases the risk of vascular occlusions and ischemic events, promoting thrombosis and neuronal death. Over time, these changes can lead to chronic cerebrovascular diseases, like small vessel disease, which contribute to neurodegeneration [207,208,209].

Additionally, the compromised cerebral microvasculature promotes cerebral microbleeds [210]. Chronic hypertension weakens the microvasculature, increasing the risk of rupture and haemorrhage, particularly in deep brain regions [211]. These microbleeds elevate the risk of stroke and neurodegeneration, exacerbating brain injury and inflammation and creating a cycle of vascular and neural damage [50].

In summary, these findings underscore how elevated BP creates an environment conducive to neurodegeneration, potentially initiating subsequent macroscopic changes in the brain (Figure 2).

#### 3.10.2. BP-Related Macroscopic Changes

The above microscopic changes result in macroscopic alterations across various brain regions [76,212,213]. This includes global and regional brain atrophy [76,212,213], with a notably strong impact observed in the hippocampus [76]. Importantly, the negative association between higher BP and lower brain volumes is also observed below the hypertension threshold [212,213]. Midlife high SBP and DBP above 130/80 mmHg are linked to a decrease of −0.27% and −0.62%, respectively, in whole-brain volume later in life [213]. Considering that BP tends to increase steadily with age [119,120], it may have a cumulative impact on brain health over the lifespan. However, when and the extent to which non-hypertensive but high BP contributes to brain health remains incompletely understood (Figure 2).

To address this question, we have recently performed a comprehensive systematic review including 52 studies (n = 343,794; 53.2% female; mean age = 58.7 years). We found consistent evidence indicating that higher BP levels are associated with smaller brain volumes, showing a dose-dependent relationship. For every 10 mmHg increase in SBP above 120 mmHg, there was an 11.2% increase in WML volume and a 1.3% reduction in hippocampal volume [7]. This aligns with a recent meta-analysis indicating that a one-standard-deviation increase in SBP corresponds to a 10% increase in WML volume [214]. Importantly, this effect was observed even below the hypertension and pre-hypertension threshold, suggesting that effects due to increases in BP, which would typically not be considered clinically significant, may significantly influence brain ageing over time. A notable concern is the decline in hippocampal volume; high BP may contribute to an additional 2.6% volume reduction, potentially bringing forward the onset of clinical dementia by up to one year [7]. Moreover, the negative impact of BP levels extend to GM, WM, and the amygdala, indicating BP’s comprehensive influence on overall brain health [7,76,212,213]. However, there were limited quantitative data available to conduct a meta-analysis in these regions [7]. Therefore, further high-quality research is needed to fully understand BP’s contribution to brain morphology.

The impact of BP on brain volumes exhibits an age-related variation, and midlife has been proposed as a critical period where high BP initiates metabolic and macroscopic changes which seed the processes that make a strong contribution to brain atrophy in later life [212,213,215]. However, the precise timing of these initial stages remains unclear. In contrast, at older ages, inconsistent associations are observed [212,215,216]. For instance, BP levels below 130/80 mmHg are linked to smaller brain volumes and larger WMLs in the elderly [7,212,213,215]; possibly due to inadequate cerebral perfusion and compromised nutrient delivery associated with lower BP, this vascular compromise may contribute to neurodegenerative processes. Conversely, we found that higher BP in individuals in their 60s is associated with smaller brain volume, and this association persists even into the 70s and above [7]. This is possibly due to chronic hypertensive effects which may lead to arteriosclerosis, narrowing and stiffening the brain’s blood vessels, leading to reduced blood flow and neurodegenerative changes [35].

Beyond age-related findings, considering the known cardiovascular health differences between sexes [106,217], the link between BP and brain structure may also differ based on sex. Indeed, a recent study found that BP levels are associated with a smaller hippocampus volume, and changes in white matter integrity were more pronounced in men compared to women (n = 427; 60.6% female; and aged 50) [218]. Other studies have provided evidence in the opposite direction, suggesting that higher BP may be associated with smaller regional brain volumes in women (n = 266, aged 68–73 years) [219]. When combining the existing evidence in our systematic review, weak sex-based differences were observed, though this was potentially due to methodological factors such as analyses being controlled for sex rather than conducted separately and stratified by sex [7]. This highlighted the need for high-quality research to investigate this question with larger sample sizes.

To address this question, we recently examined the associations between BP levels and brain volumes, as well as WMLs, in a large cohort from the United Kingdom (UK) Biobank study (n = 36,000; female 54.2%; aged 37–73 years). We carefully stratified the sample by sex and age groups (≤45, 46–55, 56–65, and >65 years), and investigated interactions with body mass index (BMI) and antihypertensive medication use [8]. In line with previous research [7], we confirmed that high BP is associated with lower brain volumes and larger WMLs and these associations were evident across all BP levels, including within the normal range [8]. Importantly, the deleterious effects of BP levels were apparent in all age groups, with a more pronounced association in 45 year olds with hypertension. This group exhibited a 0.7% lower GM and a 0.8% larger WMLs volume, equivalent to 1–2 years of typical ageing [220]. Moreover, we demonstrated sex-related differences, with a more noticeable adverse impact in women compared to men, suggesting women are more vulnerable to the detrimental effects of elevated BP than men. This may be attributable to the regulatory role of sex hormones, especially oestrogen, in BP levels because oestrogen influences vascular tone and endothelial function [106,221], potentially impacting BP regulation in women. Importantly, given the higher dementia prevalence in women globally, these findings may suggest that closely monitoring BP levels in women may be particularly beneficial and may help decrease dementia prevalence in this group.

Additionally, we found that the detrimental impact of BP on brain health is exacerbated by higher BMI. Indeed, overweight or obese individuals experienced an additional ~0.4% reduction in GM volume for every decade above 45 years, suggesting the importance of maintaining a normal weight to mitigate high BP-related pathologies [149,222]. This aligns with research showing that higher BMI is associated with lower brain and hippocampal volumes [223]. This might be because high BMI is associated with increased inflammation, and OS, and therefore contributes to cardiovascular disease and the brain’s ageing process [224,225].

Interestingly, we also found that the use of antihypertensive medication can mitigate the negative impact of BP on neurodegeneration and that it provides some neuroprotection. SBP in participants taking antihypertensive medication was associated with (−1.2%) lower WMLs in older individuals > 65 years [8]. This finding is in line with a meta-analysis of seven studies including individuals aged 60 to 78 years, showing that maintaining SBP within the 110 to 129 mmHg range with antihypertensive treatment is linked to a significant reduction (SMD = −0.37 cm^3^) in WML progression, compared to controls [226]. Furthermore, we found that antihypertensive medication appears to preserve GM, WM, and hippocampal volume in younger individuals ≤ 45 years [8]. This protective effect may be attributed to the medication’s beneficial impact on reducing inflammation, improving vascular health, thereby controlling BP. This, in turn, enhances cerebral blood flow and mitigates neurodegeneration. Certain antihypertensives also modulate neurotransmitter activity, indirectly influencing brain health [227,228]. However, contradictory evidence indicates a lack of such a protective effect in older individuals regarding brain shrinkage [226]. This may suggest that initiation of antihypertensive treatment at a younger age may be more effective in shielding the brain from the detrimental effects of elevated BP levels and potentially in reducing the future risk of dementia (Figure 2).

### 3.11. BP and Cognitive Decline

The BP-related cerebral changes described in the previous section would be expected to impact brain function and, indeed, robust evidence indicates that elevated BP seems to impact cognitive performance across various domains in individuals without dementia (Table 1) [9,229].

This influence encompasses executive function, memory, motor speed, attention, and varying degrees of cognitive decline, with particular significance in the elderly population [9,239]. The observed association follows a dose-dependent relationship [13,243]. For instance, findings from the Honolulu–Asia Aging Study (n = 3734; mean age 78 years) revealed that a 10 mmHg increase in SBP over approximately 20 years is linked to a 9% higher risk of cognitive decline [243]. Such an effect of high BP in midlife could lead to a 39% increased risk of developing dementia over about 25 years. Notably, this risk extends even below the hypertensive range (BP ≥ 120/80 mmHg and <140/90 mmHg), with a 31% increased risk of dementia [241] (Figure 2).

Given higher BP is associated with lower cognitive performance, it may be expected that early treatment with antihypertensive medication would mitigate this effect, and this is indeed what has been reported in the literature. Summary evidence from a recent meta-analysis showed that antihypertensive drugs effectively lower BP and decrease the likelihood of cognitive decline in individuals with hypertension [248]. Particularly, these medications are associated with cognitive improvements in memory and attention, with a less pronounced effect on the speed of processing or executive function [248]. Antihypertensive medication also reduces the risk of dementia. In a meta-analysis of six studies (n = 31,000; age 55 years), controlling BP through medication was linked to a 12% decreased risk of dementia and a 16% lower risk of AD over 7 to 22 years of follow-ups [249]. However, the protective effects of antihypertensive medication seems to diminish in advanced age. This observation is supported by a meta-analysis of five RCTs involving somewhat older individuals (average age 65.7 years; age range 63.0 to 80.5 years; n = 17,396; 40% female), which revealed no significant association with the onset of cognitive decline, MCI and dementia [250]. This aligns with the broader evidence suggesting a more robust protective effect of antihypertensive medication against neurodegeneration at younger ages (Section 3.10). This could be attributed to the limited effectiveness of BP medication on the brain when damage is already established. This supports the notion that damage occurs in mid-life or earlier. Therefore, early pharmacological intervention, ideally before hypertension develops, could significantly contribute to protecting the brain and reducing the dementia burden in the population. Since the evidence in the previous section suggests that the negative impact of BP is observed even in the non-hypertensive range, where individuals may not be eligible for pharmaceutical interventions [78,251]. Thus, there is a need for non-medicalized and highly scalable interventions applicable to a younger population. In the following section, we will explore whether Mg supplementation may contribute to such interventions.

## 4. Dietary Magnesium: Underlying Mechanisms and Possible Prevention Opportunity

Mg is an essential mineral that plays an important role in the ageing process. It has been closely associated with improving vascular health, notably by diminishing vascular atherosclerosis and calcification [252]. Recent meta-analyses of prospective cohort studies and randomized controlled trials have consistently demonstrated a robust inverse relationship between higher Mg intake and lower BP levels, as well as a decreased risk of developing hypertension [161]. Beyond cardiovascular health, the importance of Mg extends to brain health with several studies showing an association between higher Mg levels and reduced brain lesions [253]. Additionally, increased Mg levels have been linked to enhanced cognitive function and reduced risk of dementia [254,255,256,257], making it an important element to consider in the context of overall brain health, since cardiovascular health and high BP are important risk factors for neurodegeneration [7,8], cognitive decline [9], and dementia [11]. Mg supplementation either through pharmaceutical preparations or through dietary intake could be used in the population to lower BP levels and consequently improve brain health. However, the precise contribution dietary Mg can make in relation to brain health remains unclear and requires further investigation. The following section will explore the potential role of Mg in moderating the ageing process, focusing on its impact on BP regulation and its potential influence on brain health.

### 4.1. Magnesium and Ageing Mechanisms

Research has revealed a strong link between Mg intake and the mechanical properties of blood vessels [252]. In an animal study, rats fed a Mg-deficient diet exhibited thicker intima-media layers within their blood vessels compared to rats on a Mg-supplemented diet [258]. A reduction in the carotid intima-media thickness (CIMT) was also demonstrated after a 24-week course of Mg supplementation (250 mg/day) in diabetic haemodialysis patients, compared to a placebo (*p* = 0.004) [160]. This observed improvement in CIMT could be attributed to Mg’s role in preventing the development of atherosclerotic plaques [259,260,261]. A 6-month Mg supplementation (600 mg/day) significantly reduced SBP in hypertensive women. Interestingly, the placebo group experienced an increase in CIMT, indicating atherosclerosis progression [259]. The most likely mechanism is the role of Mg in reducing lipid build-up in arterial walls [262]. Robust evidence indicates a notable reduction in serum LDL levels (*p* = 0.006) with Mg supplementation, especially at doses exceeding 300 mg/day, in individuals with type 2 diabetes. Additionally, a positive association was observed between higher Mg intake and an increase in HDL levels (*p* = 0.026) [263].

Mg influences vascular contraction by regulating Ca levels, through a mild physiologic Ca channel blocker action [264]. This action results in a decrease in both cardiac [265] and aortic contractility [266]. Importantly, the regulatory influence of Mg extends to preventing excessive Ca accumulation in arteries, thereby reducing the risk of vascular calcification [266,267]. This effect was shown in the Framingham Heart Study (n = 2695; mean age 54 years), revealing that higher dietary Mg intake (427 mg/day) is associated with a 58% lower coronary artery calcium (CAC) in individuals without cardiovascular disease compared to those with lower intake (258 mg/day) [268].

Importantly, these vascular effects contribute directly to lowering BP and reducing the risk of hypertension. In a spontaneously hypertensive rat model, lower Mg serum and tissue levels were observed [269,270]. Furthermore, Mg supplementation (650 mg/day for 10 weeks) was shown to lower BP in young prehypertensive rats, but had no effect in older rats [270]. Consistent findings from meta-analysis studies in humans have also highlighted the effectiveness of Mg supplementation in reducing BP levels and preventing the development of hypertension [161]. Indeed, Mg administration at doses ranging from 365 to 450 mg/day led to a substantial reduction of 4.18 mmHg in SBP, and 2.18 mmHg in DBP with follow-ups ranging from 1 to 6 months [17]. Notably, these positive effects were observed in individuals with insulin resistance or type 2 diabetes, suggesting the broad applicability and potential benefits of Mg supplementation across diverse populations and conditions [17]. Research indicates that even a slight decrease in BP is clinically significant in lowering the risk of coronary heart disease and stroke [271]. Furthermore, meta-analysis of 34 trials involving diverse populations found an inverse relationship between Mg supplementation and BP [272]. Paradoxically, higher serum Mg concentration have not been found to be associated with a reduced risk of hypertension [16]. However, since Mg is primarily stored intracellularly, with only ~1% circulating in the bloodstream, it is likely that serum Mg is a poor index of Mg intake and is most useful in detecting major Mg deficiency [15]. This underscores the importance of focusing on dietary intake and supplementation to assess Mg status more accurately.

To better understand the clinical significance of Mg supplementation, it is important to consider the maximum potential increase in plasma Mg levels. Normal plasma Mg levels typically range from 1.7 to 2.4 mg/dL. Toxicity symptoms have been reported at plasma Mg levels between 7 and 12 mg/dL, and cardiorespiratory arrest may occur at levels exceeding 15 mg/dL [273]. While supplementation can raise plasma Mg levels, these increases are usually small; for example, a recent meta-analysis including 41 RCT showed that a median dose of 365 mg per day over a median period of 12 weeks, serum Mg levels increased by only 0.12 mg/dL [274]. This underscores that while Mg supplementation leads to only modest increases in plasma levels, its clinical benefits may still be significant due to its effects on cellular and metabolic processes.

In addition to direct vascular effects, Mg protects the brain from oxidative stress by stabilizing mitochondrial membranes and supporting antioxidant enzymes, such as glutathione peroxidase and superoxide dismutase (SOD), which neutralize ROS. This preserves neuronal integrity and function, especially in ageing populations [275]. Mg may contribute to improve brain health through anti-inflammatory effects [276]. Indeed, low Mg levels promote microglia activation, and the release of IL-6, TNF-α, and nitric oxide [277]. In contrast, Mg supplementation in rats has been found to reduce microglia activation and inhibit TNF-α production [278]. Similarly, in humans, a recent meta-analysis across 15 RCTs (n = 889 participants; females = 62.5%; mean age 46 years) revealed that Mg supplementation compared to placebo significantly lowered CRP blood levels (SMD = −0.356, 95% CI −1.224, 0.017). Mg supplementation also significantly reduced plasma fibrinogen, TNF, and IL-1 [279]. Notably, Asbaghi et al. [263] suggested a link between higher Mg levels and increased HDL levels (*p* = 0.026), potentially offering cardiovascular benefits by increasing the anti-inflammatory effect of HDL [280] (Figure 3).

In addition, Mg helps maintain the integrity of the BBB [281]. Low Mg levels have been associated with BBB dysfunction, resulting in increased permeability and allowing neurotoxic substances to enter the brain [281,282]. Mg also contributes to maintaining neuronal function. It is essential in regulating the activity of N-methyl-D-aspartate (NMDA) receptors, which are important for synaptic plasticity and memory function. Excessive activation of NMDA receptors leads to an influx of Ca ions into neurons, resulting in excitotoxicity and potential neuronal damage. Mg acts as a natural NMDA receptor antagonist, blocking excessive Ca influx and thereby protecting against synaptic loss and maintaining synaptic function [263,283]. Inadequate Mg levels can result in the overactivity of NMDA receptors, increasing Ca flow into neurons and leading to synaptic loss and neuronal damage [284].

Finally, Mg acts as a cofactor in key enzymatic reactions for neurotransmitter synthesis, including serotonin, dopamine, and gamma-aminobutyric acid (GABA) [285]. By regulating these systems, Mg significantly influences mood, cognition, and stress responses. Therefore, sufficient Mg levels are important for GABAergic activity, which helps prevent anxiety and depression [286].

### 4.2. Magnesium and Brain Ageing

Glick [287] was amongst the first to hypothesise a protective effect of Mg against age-related neurodegeneration. Subsequent research has explored underlying mechanisms for this protective effect, including the important role of Mg ions in neuronal maturation, and their physiological presence in the cerebrospinal fluid (CSF) and within brain tissue [288]. This prompted several lines of research to investigate the potential benefits of Mg supplementation.

In rats supplemented with Mg through diet, a 15% increase in CSF Mg levels and an approximately 30% rise in the total Mg concentration was observed in the brain tissue [289]. This increase was shown to contribute to the protection of neuronal function, the prevention of synaptic loss [289,290], and the proliferation of neuronal stem cells (NSC) in the hippocampus, ultimately enhancing neurogenesis [291,292]. Importantly, these beneficial effects on NSC was found to occur in both young and aged mice [292]. Additionally, high Mg levels in the brain were found to be associated with a decrease in glial activation and neuroinflammation [293,294,295]. This has been found to be protective for myelin and white matter fibres [296,297], thereby maintaining good signal transmission between neurons and improving learning and memory abilities [298]. In addition, Mg supplementation was found to reduce Aβ accumulation [289,295,299], and tau hyperphosphorylation which [253], given their role in AD and other dementias, is likely to contribute to a reduced risk of developing these conditions.

Converging evidence consistent with Mg’s neuroprotective effects was also demonstrated in humans, with accumulating evidence consistently demonstrating that serum Mg levels appear to be associated with dementia. For example, lower Mg plasma concentrations [300] in T-cells, blood cells [301], CSF [302,303], and hair [304,305] were reported in individuals with AD. These variations in Mg levels across different biological markers indicate a potential link between Mg and neurodegeneration. Post-mortem histological analyses have consistently shown lower Mg levels in AD brains compared to those of healthy controls [287,305], with differences being particularly salient in the hippocampus [306]. While these observations consistently show lower Mg levels in neurodegenerative diseases, it is crucial to recognize that these biological levels primarily reflect the current Mg state. To achieve a more comprehensive understanding of Mg’s neuroprotective effects, investigating dietary intake becomes imperative. This approach helps bridge the gap between biological markers and dietary considerations, providing insights into the intricate relationship between Mg and brain health. However, a gap in knowledge exists concerning the impact of Mg intake on neurodegeneration (Figure 3).

To address this, we recently investigated the associations between dietary Mg intake and brain volumes, as well as WMLs, in a large cohort of cognitively healthy individuals (n = 6001; 54.7% female; age range 40–70 years). Our findings showed that individuals with a higher Mg intake (550 mg/day or more) had a 0.20% larger GM and a 0.46% larger right hippocampal volume compared to those with a typical Mg intake of around 350 mg per day [307]. This is a strong effect as this volumetric difference equates to approximately one additional year of brain ageing [59]. These effects are particularly noteworthy because they were observed in a general population with normal Mg levels (mean 355.35 mg), and therefore are not attributable to Mg deficiency. Thus, if these results extend to other populations, a 41% increase in Mg intake could potentially have a substantial positive impact on brain health and may contribute to preserving cognitive abilities and to reducing the risk or delaying the onset of dementia [307] (Figure 3).

Interestingly, such effects have also been demonstrated in relation to cognitive function and decreased dementia risk in epidemiological studies [308]. For example, in a Japanese population, individuals with the highest Mg dietary intake (≥196 mg/day) exhibited a 37% lower risk of all types of dementia compared to those with the lowest intake (≤174 mg/day) in a large population (n = 1000; mean age 69 years) over a 17-year follow-up period [255]. This trend is consistent with another population study (n = 1406, 52% female, mean age 62.5 years) where higher Mg intake (≥434 mg) was associated with a 93% reduced risk of transitioning to MCI [257]. Importantly, these benefits extend beyond clinical outcomes, as higher Mg intake has also been linked to improved cognitive function in unimpaired individuals [256]. Together, these findings suggest that Mg supplementation may have broad benefits for cognitive health in mid to late life. However, a major knowledge gap pertains to whether such effects can also be demonstrated at younger ages.

To address this question, we investigated the links between dietary Mg intake and both brain volumes and WMLs in individuals aged 40 to 70 years. We found that the protective effect of dietary Mg emerges at least as early as the 40s, and perhaps earlier [307]. This observation may be attributable to a broader body of evidence suggesting that factors influencing both the risk and protection against brain ageing emerge in early life and accumulate throughout the entire lifespan [204]. This aligns with the observations of Ozawa et al. [255] and Lo et al. [254] who identified a positive relationship between dietary Mg intake in middle age and the onset of dementia 17 to 20 years later. This suggests that Mg is a crucial intervention for preserving and promoting brain health in the general population, with a preference for implementation in younger age cohorts.

Beside age, the neuroprotective effects of dietary Mg also vary by sex. Tao et al. [256] found a stronger positive association between high Mg intake and higher cognitive scores in women compared to men. Similarly, we also observed a stronger relationships between high Mg intake, larger brain volumes, and lower WMLs in women compared to men [307]. This may be attributable to hormonal factors, as oestrogen has been found to be protective against neurodegeneration, and dementia [309]. To explore this hormonal hypothesis, we contrasted pre- and post-menopausal women and contrary to our hypothesis, we observed a stronger positive association between Mg intake and brain volumes in post-menopausal women across various brain regions, as compared to pre-menopausal women [307]. Thus, an alternative hypothesis is that the observed effect may be due to the anti-inflammatory effects of Mg, as Chacko et al. [310] previously found lower inflammatory markers in post-menopausal women with a higher dietary Mg intake. These are important findings because it has been widely demonstrated that post-menopausal women experience more neuroinflammation and are exposed to more cardio-metabolic risk factors than pre-menopausal women, and therefore may benefit more from Mg’s anti-inflammatory effects.

In summary, consistent evidence suggests that Mg has neuroprotective effects (Table 2). This makes dietary Mg a potential agent for promoting brain health. However, the precise mechanisms responsible for this protective effect are not fully understood and are explored in the following section.

### 4.3. Mechanisms Mediating Magnesium Effect on Brain Ageing/Cognitive Function

Relatively little research directly investigating the mechanism underlying the neuroprotective effects of dietary Mg has been conducted. It is well-established that high BP levels contribute to neurodegeneration [7,8,329], cognitive decline [9], and increase the risk of dementia [11]. Since increased Mg intake decreases BP levels, a logical hypothesis is that the association between higher Mg levels and reduced neurodegeneration, improved cognitive function, and a lowered risk of dementia is attributable to its effect on BP. Surprisingly, in a recent study testing this hypothesis, we did not find evidence supporting such a mediation effect [307].

Thus, an alternative hypothesis is that since, as reviewed in Section 3.10.1, Mg’s beneficial mechanisms also include downregulating the inflammatory response, it is possible that Mg’s anti-inflammatory effect mediates its neuroprotective effect. We recently tested this hypothesis and found some evidence that inflammation mediates Mg’s neuroprotective effect. Specifically, higher Mg intake was associated with lower levels of high-sensitivity CRP and this effect was found to partly explain the identified association between higher dietary Mg intake and larger GM volume [330]. It is important to note that this anti-inflammatory effect partially explained, but did not fully mediate, the Mg effect on brain volumes. Therefore, another alternative mechanism may also be involved. This may include an antihypertensive effect that may have occurred in the years or decades before as the quality of dietary intake tends to remain relatively similar in adulthood [331]. Moreover, as suggested in Section 3.10.1, Mg may also potentially contribute to neuroprotection by blocking NMDA receptors, and thus preventing some synaptic loss, and maintaining synaptic function [254,283]. However, there is limited evidence available on these associations in humans. Further studies are needed to explore and validate these potential mechanisms, which may offer a more comprehensive understanding of Mg’s role in preserving brain health (Figure 4).

## 5. Conclusions

As a whole, the body of evidence presented above strongly suggests that high BP levels are associated with neurodegeneration and cerebrovascular disease, cognitive decline, and the development of dementia, and ultimately, premature death (Table 1). A key finding is that these effects appear to occur across the entire BP spectrum and are observable even in young individuals. This is highly relevant to population health as it suggests that relatively small increases in BP across the whole life course may contribute to a substantial disease burden.

Another important finding indicates that Mg intake promotes neuronal health, protects against neurodegeneration, enhances cognitive function, and lowers the risk of dementia, particularly among younger people (Table 2). Thus, incorporating and encouraging high Mg intake in our diet or through supplements, beginning at a young age and across the lifespan, represents a simple strategy to improve cognitive function and reduce the risk of dementia in the population and should be considered as a focus of scalable population health interventions.

An unexpected finding also emerged from this review. Our initial hypothesis postulated that the predicted, and eventually confirmed, associations between Mg and brain volumes would be mediated by a decrease in BP attributable to Mg’s widely demonstrated antihypertensive effect. Intriguingly and contrary to expectations, we found no direct evidence for such a mediation effect in the human population studied. Instead, some limited evidence suggesting that reduced inflammation associated with Mg intake might mediate its effect on brain health was identified. Nonetheless, it is still more likely than not, based on the overall evidence in the literature, that the BP-lowering effect of Mg contributes to better brain health. However, further research aimed at more specifically confirming this is needed.

An important new insight from this review that needs to be further explored in future research is that the anti-inflammatory action of Mg may play a role in promoting brain health in humans. This is consistent with the limited converging animal evidence on this topic, but future research should seek to more definitely address this question as it may have important implications for chronic disease prevention and healthy lifestyle messaging. Finally, this review has also identified a potential but under-researched alternative neuroprotective mechanism whereby Mg may block NMDA-induced cytotoxicity, consequently reducing synapse loss and preserving synaptic function which requires more attention.

In summary, this review contributes to the growing body of knowledge on the complex interplay between BP levels, dietary Mg intake, inflammation, and its implication for brain health. Since these effects are active across the lifespan, they present substantial opportunities to implement risk-reduction strategies in younger individuals, which have the potential to lower disease burden in the population over decades.

## Figures and Tables

**Figure 1 ijms-25-11859-f001:**
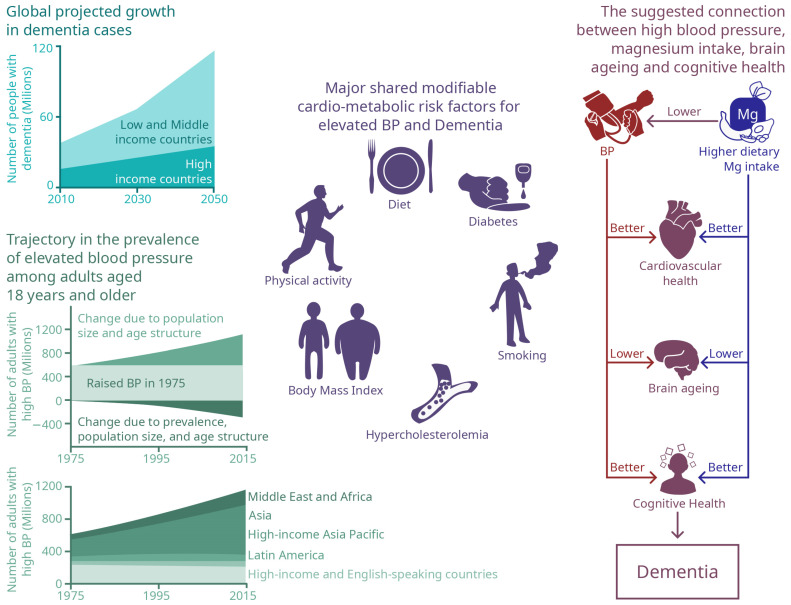
The figure shows the global rise in brain ageing cases and the increasing incidence of dementia, highlighting the urgent need for prevention. A major risk factor is high blood pressure (BP), which shares several common risk factors with brain ageing. High BP affects a significant portion of the population worldwide, with recent findings indicating an earlier onset of hypertension in younger individuals. Higher magnesium intake is associated with reduced BP and improved cardiovascular health. Additionally, the protective effects of magnesium extend to lowering the risk of cognitive decline and brain ageing.

**Figure 2 ijms-25-11859-f002:**
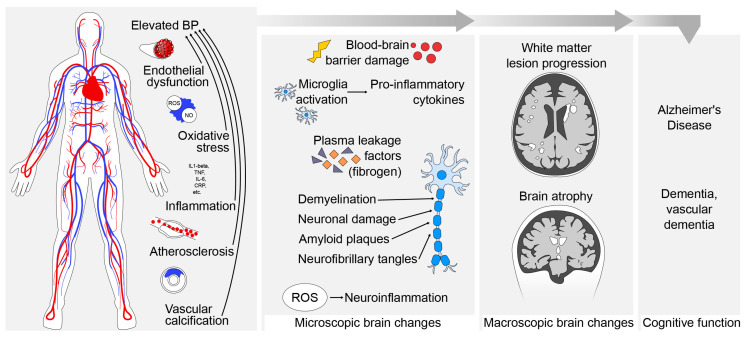
Schematic overview of the relationship between blood pressure (BP), ageing mechanisms, brain changes, and cognitive function. It depicts how increased BP adversely affects endothelial function, leading to oxidative stress, inflammation, vascular atherosclerosis, and calcification, which may further elevate BP. Chronically high BP is associated with microscopic changes in the brain, including blood–brain barrier (BBB) disruption, microglia activation, pro-inflammatory responses, demyelination, and the accumulation of amyloid plaques and tau protein. These microscopic alterations result in macroscopic changes such as larger white matter lesions and reduced brain size, ultimately contributing to accelerated brain ageing and an increased risk of dementia.

**Figure 3 ijms-25-11859-f003:**
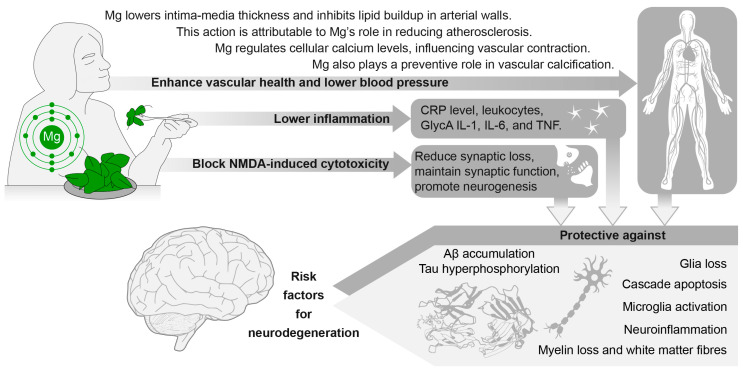
Schematic overview illustrating the relationship between magnesium (Mg), ageing mechanisms, and micro- and macrostructural brain changes. It shows how higher dietary Mg intake supports vascular health by improving endothelial function and regulating blood pressure (BP). Mg also contributes to reducing inflammatory processes, thereby mitigating the harmful effects of chronic inflammation on brain tissue. Additionally, Mg exhibits neuroprotective properties, such as the ability to block NMDAR (N-methyl-D-aspartate receptors), which is involved in excitotoxicity—a process that can damage neurons during ageing. These beneficial effects of Mg lead to a reduction in various brain-ageing pathologies, including glial cell loss, microglia activation, oxidative stress, neuroinflammation, demyelination, amyloid accumulation, and tau phosphorylation. By counteracting these pathological mechanisms, Mg intake may help preserve brain health and slow down age-related brain degeneration, ultimately reducing the risk of cognitive decline.

**Figure 4 ijms-25-11859-f004:**
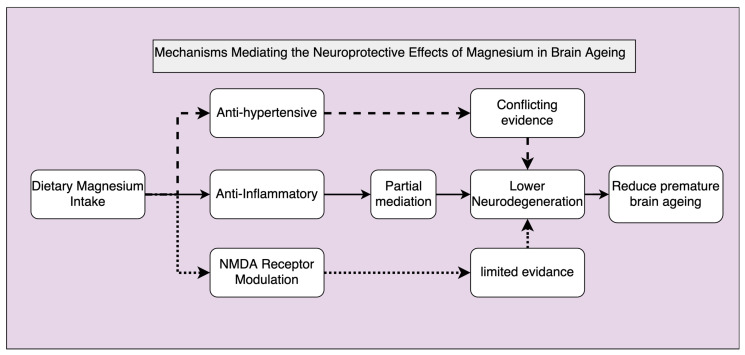
Schematic overview of mechanisms mediating the neuroprotective effects of magnesium on brain ageing. The antihypertensive effect of magnesium may be linked to slower ageing, although recent research by Alateeq et al. [307] found no significant effects in a large population sample. Emerging evidence suggests that inflammation may partially mediate the association between magnesium intake and brain volume, indicating that the anti-inflammatory properties of magnesium could help reduce brain ageing. Furthermore, multiple pathways, such as NMDA receptor modulation, may play a role in the neuroprotective effects of magnesium. While evidence from human studies on NMDA modulation remains limited, further research is needed to fully elucidate these interconnected mechanisms.

**Table 1 ijms-25-11859-t001:** Association between higher BP and ageing mechanisms, neurodegeneration, cognitive decline, and dementia.

Domain	Type of Research	Effects of ↑ BP	Effects of ↑ SBP	Effects of ↑ DBP	Effects of ↑ MAP
Ageing mechanisms	Animal	↑ Endothelium dysfunction [168,169,170].			
		↑ Proinflammatory cytokines, including IL-1β [174], TNF-α, IL-6 [175,176], and IFN-γ [177], IL-17 [180].			
		↑ Plaque formation and atherosclerosis [186,187]			
		↑ Calcium deposits and vessel stiffness [188,191,192].			
	Human	↑ Endothelial function assessed by ↑ FMD [230].			
		↑ Pro-inflammatory cytokines including IL-6 and CRP [173,182,231,232].			
		↑ Atherosclerosis [233,234].			
		↑ Calcium deposits and vessel stiffness [114].	↑ Calcium deposits and vessel stiffness [235].		
Neurodegeneration	Animal	↑ Glial and dendritic loss [18,19,236].			
		↑ Myelin disruption [38].			
		↑ Vascular impairment including ischemia [207,208,209].			
		↓ Synaptic density [202].			
		↑ Aβ-plaque deposition [204]. ↑ Tauopathy [204].			
	Human	↓ Brain volume, hippocampal volume [212,213]. ↑ WML volume [7,214].	↓ GM, WM, hippocampal volume [7,212,213,237], and amygdala volume [237]. ↑ WML volume [7,214].	↓ GM, WM, hippocampal volume [7,212,213,237] and amygdala [237]. ↑ WML volume [7,214].	↓ GM, WM, hippocampal volume [7,237], and amygdala [237]. ↑ WML volume [7].
Cognitive decline	Animal	↑ Cognitive decline [238].			
	Human	↓ Executive function, memory, Motor speed, and attention [9,239]. ↑ Cognitive decline [240,240,241,242].	↑ Cognitive decline [242,243].		
Dementia	Animal	↑ AD like pathology [244].			
	Human	↑ Vascular dementia risk [245]. ↑ AD [246].	↑ Dementia risk [241,247]. ↑ AD [247].		

**Abbreviation**: AD: Alzheimer’s disease; Aβ-plaque: Beta-amyloid plaque; CRP: C-reactive protein; FMD: flow-mediated dilation; GM: gray matter; IL: interleukin; WM: white matter. ↑: higher/increase; and ↓: lower/decrease. Colour reference: Green for aging mechanisms, pink for neurodegeneration, gray for cognitive decline, and blue for dementia incidence, with light shades for animal studies and dark shades for human studies.

**Table 2 ijms-25-11859-t002:** Association between magnesium and ageing mechanisms, neurodegeneration, cognitive decline, and dementia.

Domain	Type of Research	Effects of ↑ Mg (Serum)	Effects of ↑ Mg (Dietary)	Effects of ↑ Mg (Brain Levels)
Ageing mechanisms	Animal		↓ Thickness of intima-media layers in blood vessels [258].	
			↓ Atherosclerosis [311,312].	
			↓ Plasma oxLDL	
			↓ Vascular calcification [313]	
		↓ BP levels [269,270].	↓ BP levels [269].	
			↓ Expression of proinflammatory cytokines including TNF-a and IL-1B [294].	↓ Microglia activation inhibition [314,315]. ↓ Expression of pro-inflammatory cytokines, including TNF-α, IL-1α, IL-1β, and IL-6 [295,314,315].
				↓ Blocking of cytotoxic effects of NMDA [283].
	Human	↓ Carotid intima-media thickness [316].	↓ Carotid intima-media thickness [160].	
		↓ Atherosclerotic plaques [317,318].	↓ Atherosclerotic plaques [259,261].	
		↓ LDL levels [319,320,321,322].	↓ Serum LDL levels [263]. ↑ HDL levels [263].	
			↓ Coronary artery calcium [268]. No effect on vascular calcification [323].	
		No link between serum Mg concentration and reduced risk of hypertension [16].	↓ BP (e.g., SBP, DBP) levels [16,17,324].	
		↓ Chronic inflammation [276]. No link between serum Mg and inflammation markers including CRP and ESR [316].	↓ CRP levels [278,279,325] ↓ Blood levels of pro-inflammatory cytokines, including IL-6 and TNF-α [278,325].	
Neurodegeneration	Animal		↑ NSC [292].	
			↓ Synaptic loss [289,298,326].	↑ Synaptic plasticity [253]
			↓ Aβ-plaque deposition [289,293].	↓ Aβ-plaque deposition [289,295]. ↓ Tau hyperphosphorylation [253].
	Human		↑ Brain volumes including GM, WM, and hippocampal volume [307]. ↓ WMLs [307].	
Cognitive decline	Animal		↑ Learning and memory abilities [289,298].	↑ Learning and memory abilities [253].
	Human	↑ Cognitive function [327].	↓ Cognitive decline [256]. ↓ Transition to MCI [254,257].	
Dementia	Animal		↑ Cognitive function in AD mic [298].	
	Human	↓ Plasma Mg in AD [300]. No association between plasma Mg concentrations and AD [328].	↓ Dementia [254]. ↓ Vascular dementia and AD and all Dementia type [255].	

**Abbreviation:** Aβ-plaque: Beta amyloid plaque; AD: Alzheimer’s disease; BP: blood pressure; CRP—C-reactive protein; DBP: diastolic blood pressure; ESR: Erythrocyte sedimentation rate; GM: Gray matter; HDL: high-density lipoprotein; IL: interleukin; MCI: mild cognitive impairment; NMDA: N-methyl D aspartate; NSC: neural stem cells; oxLDL: oxidized low density lipoprotein; SBP: systolic blood pressure; TNF-α: tumour necrosis factor-alpha; WMLs: white matter lesions; WM: white matter. ↑: higher/increase; and ↓: lower/decrease. Colour reference: Green for aging mechanisms, pink for neurodegeneration, gray for cognitive decline, and blue for dementia incidence, with light shades for animal studies and dark shades for human studies.

## Data Availability

Not applicable.

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
