# Peer review of "High Blood Pressure and Impaired Brain Health: Investigating the Neuroprotective Potential of Magnesium"

_ijms, 2024, doi:10.3390/ijms252211859_

Round 1
Reviewer 1 Report (New Reviewer)
Comments and Suggestions for Authors Authors have chosen a clinically very relevant topic. High blood pressure is a crucial problem in health care worldwide. Untreated or uninfluenced high blood pressure can lead to many pathological sequels. Among others, high blood pressure can be related to neurodegeneration, cerebrovascular disease, cognitive decline, dementia and so to premature death. The manuscript discusses every aspect of high blood pressure, ageing and their effect on brain health and integrity. A potential useful supplement has been added to the review about hogh blood pressure and brain health, the magnesium. In the Introduction section the authors explain the basic aim of the review and then in the following sections we can get a general insight into the physiological mechanisms of ageing, regarding microscopic and macroscopic changes of the brain, cognitive decline. In the third section authors describe the relation between blood pressure and ageing. The authors give a detailed explanation of high blood pressure, regulatory mechanisms of blood pressure, measurement, epidemiology, risk factors and treatment. Besides, in 3.9 and 3.10. parts of the manuscript summarize the available information on blood pressure and ageing and blood pressure and brain ageing, respectively. In Fig. 1. the connection between blood pressure and ageing, brain damage, cognitive decline is demonstrated. In addition, Table summarizes the association between higher blood pressure and ageing mechanisms, neurodegeneration, cognitive decline and dementia. Authors have shared every in the literature available information about the connection between Mg and high blood pressure and brain health. This review summarizes the possible effect of Mg on the high blood pressure and its sequels. Fig. 2. illustrates the relationship between Mg, ageing mechanisms and brain changes. Table 2. summarizes the effects of Mg on ageing mechnism, neurodegeneration, cognitive decline and dementia. At the end, we can read a conclusion summarizing the connection between Mg and bran health. The manuscript is well-written, detailled enough. The mannuscript is built-up systematically. The sections following the ones before explain the connections between blood pressure, ageing and brain changes. The conclusions are supported by the cited literature. At the end of the manuscript, an approprate list of references can be read. I have noticed only one mistake on page 6 line 257: the end of the sentence is missing. In summary, the manuscript can be accepted in its present form.Author Response
Dear Editor,
Please find attached our cover letter and response to the reviewer’s comments.
Best regards,

Reviewer 2 Report (New Reviewer)
Comments and Suggestions for Authors
In the manuscript " High Blood Pressure and impaired Brain Health: Investigating the Neuroprotective Potential of Magnesium, Alateeq et al reviewed the possible role of Mg in neuroprotection. This manuscript is of scientific and practical interest, and relatively well structured and written. They revised their manuscript significantly; however, the authors should consider provide more figures to fully summarize any piece of this article. The figure legend should be stated more. Other review papers in this topic should be cited. Overall, this review article is not yet ready for publication.
Author Response
Dear Editor,
Please find attached our cover letter and response to the reviewer’s comments.
Best regards,

Round 2
Reviewer 2 Report (New Reviewer)
Comments and Suggestions for Authors
Accept in present form
This manuscript is a resubmission of an earlier submission. The following is a list of the peer review reports and author responses from that submission.
Round 1
Reviewer 1 Report
Comments and Suggestions for Authors
This manuscript explores the relationship between hypertension and impaired brain health, and discusses the potential and mechanisms of magnesium treatment. However, the article is overly lengthy, incorporating much content not directly related to the core theme. I recommend further revisions.
Comments:
1. The article intends to address hypertension and impaired brain health but extensively discusses aging and its mechanisms. While somewhat related to the theme, the detailed exposition on aging is unnecessary that long.
2. Section 3 devotes excessive pages to discussing BP's definition and regulation (essentially textbook content), the Effect of Sex Hormones, Measurement (this paper is not a guide on measuring blood pressure), and epidemiology (which could be briefly mentioned without needing an entire page).
3. Section 3.8 on BP treatment could briefly include content on the use of magnesium for treating high blood pressure, aligning it more closely with the article's main subject.
4. Similarly, Section 3.9, BP and Aging Mechanisms, strays from the article's focal point. This topic has been extensively covered in other literature. Discussing everything makes the article lengthy and unfocused. I recommend deleting this section and directly transitioning to the main topic—High Blood Pressure and Impaired Brain Health.
5. In Figure 1, the graph on the far left appears to establish a relationship chain from endothelial dysfunction → oxidative stress → inflammation → atherosclerosis. However, this depiction is not recommended. It would be more appropriate to show that these factors are all related to hypertension, as the proposed relationship chain requires more rigorous scientific evidence for substantiation. Also, the "Microscopic brain changes" in the diagram should be centered.
6. Section 3.10.1 should be more thoroughly elaborated and summarized. This section aims to discuss the potential mechanisms by which hypertension can damage brain health, which should be the focal point of this review.
7. Line 543 suggests deleting (SMD = –0.10 cm).
8. The content from Line 680-686, which the article should detail, discusses magnesium's effect on high blood pressure.
9. From a clinical benefit perspective, the article should briefly cover the maximum potential increase in Mg levels in human plasma, rather than suggesting unilateral increases.
Reviewer 2 Report
Comments and Suggestions for Authors
This review by Khawlah Alateeq et al. is interesting, suggesting high blood pressure, which is thought to play a role in the impairment of brain Health. They also suggested the role of magnesium as a capacity BP-lowering agent to promote brain health. However, although authors have reviewed aging in detail, the relationships between high blood pressure and brain health (function) have not been clearly explained in this manuscript. The author described more about inflammation and ROS generation rather than focusing on blood pressure in relation to brain health. The authors had better revise the manuscript to shorten the general descriptions of aging and blood pressure, focusing on brain health in the context of blood pressure.
Specific comments
Descriptions of general aspects of aging and their mechanism are detailed and too long. That should be summarized as concisely as possible. Instead, in this review article, authors had better concentrate on specific changes in blood pressure in the brain (blood flow) and brain aging that are closely associated with brain health and function. More importantly, I would recommend authors rather than focusing on inflammation and ROS generation, focus on blood pressure according to their relationships with brain structure and functional impairment and cognitive decline or associated diseases.
Reviewer 3 Report
Comments and Suggestions for Authors
Alateeq and their colleagues have put together a review that covers a wide range of topics related to blood pressure in systematic and brain aging and underlying mechanisms in the molecular level. Their exploration of pathophysiology effects of blood pressure and associated health risks. The benefit and role of dietary Mg in aging provides valuable insight into the area of research. Method and table sections seem to be clear and contributing to further quality of the research. The references are very relevant to the filed. It seems like this review would be a valuable resource for anyone working in the field.
Reviewer 4 Report
Comments and Suggestions for Authors
Dear Sirs,
this is a very well-written and presented manuscript. However, it could be improved if:
1. As the title refers to the usefulness of magnesium in treating hypertension and ameliorating brain health, this manuscript should be more focused upon this thesis. Therefore, I recommend mitigating the parts about hypertension, as for example the definitions of hypertension are well-known to clinicians. In addition, another example is that risk factors and epidemiology of hypertension are also well-known components of hypertension.
2. On the contrary, I recommend that the authors should add more information regarding the role of magnesium in brain health.
3. The addition of abbreviations' part is necessary. Apart from the abbreviations 'section, the authors should re-check and explain the abbreviation the first time the word is written in the text. For example, they use the term WT and they have explained it later on. The same holds true for references in the text. In some instances, the references are added as the author's name et al with year added, whereas they should appear as number at the end of the phrase. Please re-check and make any necessary changes throughout the text regarding abbreviations and references.
4. References from 2024 are lacking, whereas reference number 326 is under review??? Please add more up to date references as this is a hot matter, indeed.